# Thickness Dependence of Thermoelectric Properties and Maximum Output Power of Single Planar Sb_2_Te_3_ Films

**DOI:** 10.3390/ma15248850

**Published:** 2022-12-11

**Authors:** Prasopporn Junlabhut, Pilaipon Nuthongkum, Adul Harnwunggmoung, Pichet Limsuwan, Chanon Hatayothai, Rachsak Sakdanuphab, Aparporn Sakulkalavek

**Affiliations:** 1School of Science, King Mongkut’s Institute of Technology Ladkrabang, Chalongkrung Rd., Ladkrabang, Bangkok 10520, Thailand; 2Faculty of Science, Rajabhat Rajanagarindra University, Chachoengsao 24000, Thailand; 3Faculty of Science and Technology, Rajamangala University of Technology Suvarnabhumi, Nonthaburi 11000, Thailand; 4College of Advanced Manufacturing Innovation, King Mongkut’s Institute of Technology Ladkrabang, Chalongkrung Rd., Ladkrabang, Bangkok 10520, Thailand

**Keywords:** Sb_2_Te_3_ material, flexible film, thermoelectric property, power factor

## Abstract

P-type Sb_2_Te_3_ films with different thicknesses were deposited on polyimide substrates via heat treatment-assisted DC magnetron sputtering. The correlations between the thickness variance and the structure, dislocation density, surface morphology, thermoelectric properties and output power are investigated. As a result, it is clear that the film thickness and the heat treatment process during growth are related to the diffusion of deposited atoms on the substrate surface, leading to imperfection defects inside the films. The imperfections inside the films are affected by their properties. This work also presents the thermoelectric efficiency of a planar single leg of the deposited films with various thicknesses. The maximum power factor is 2.73 mW/mK^2^ obtained with a film thickness of 9.0 µm and an applied temperature of 100 °C. Planar Sb_2_Te_3_ produced a maximum output power of 0.032 µW for a temperature difference of 58 K.

## 1. Introduction

In recent decades, many researchers have focused on the energy harvesting from kinetic energy, thermal energy and solar or light energy, especially thermal energy harvesting has been reached for many applications [1,2,3,4]. Thermoelectric technology has been utilized for energy harvesting and reuse of waste heat. The direct conversion of waste heat energy to electrical voltage and vice versa occurs by the Seebeck and Peltier effect through thermoelectric materials. For thermoelectric generator applications, the maximum output power is required and defined as $P_{max}=\frac{S^{2}\Delta T^{2}tw}{4\rho l}$, which is proportional to the thickness (*t)* and width (*w*) of the thermoelectric leg and inversely proportional to the resistivity (*ρ*) and length (*l*). This relation indicates that the film thickness is one of the key parameters that plays an important role in the thermoelectric efficiency and generation of output power. Antimony telluride (Sb_2_Te_3_), based on the general formula $A_{2}^{V}B_{3}^{VI}$, is a narrow band gap semiconductor with a rhombohedral crystal structure and is a well-known thermoelectric material for near room temperature applications [1,2,3,4]. For wearable applications, the flexible thermoelectric materials and devices are receiving increasing attention, due to their capability to convert the heat into electricity directly by conformably attaching them onto heat sources [5,6,7,8,9,10]. Sb_2_Te_3_ thin and thick films have been deposited on flexible polymer substrates by vacuum deposition techniques such as co-evaporation [6,7,11], sputtering [8] and pulsed laser deposition [12]. Among those various techniques, magnetron sputtering is especially promising for depositing uniform and compact Sb_2_Te_3_ thin films with large area by using a single target [9,10]. However, there is another problem originating from the large lattice mismatch between the Sb_2_Te_3_ and flexible substrate, resulting in generation of a large number of dislocations [12]. In lattice mismatch systems, the electronic properties such as carrier concentration and carrier mobility are strongly thickness-dependent due to the creation of dislocations.

In this work, the Sb_2_Te_3_ films with different thicknesses deposited on polyimide substrates via thermal treatment-assisted DC magnetron sputtering were studied. The influences of film thickness on their structural and thermoelectric properties, and maximum output power of a single planar Sb_2_Te_3_ films were investigated.

## 2. Experimental Procedures

### 2.1. Flexible Sb_2_Te_3_ Film Deposition

First, polyimide (DuPont Kapton, Goodfellow Cambridge Limited, Huntingdon, England) substrates were prepared and ultrasonically cleaned, as described in our previous work [13,14]. After that, Sb_2_Te_3_ films with various deposition times of 2, 30, 60 and 120 min were deposited on flexible polyimide by DC magnetron sputtering using a single compound target (purity of 99.9%) with a diameter of 3 inches. Prior to the deposition process, the base pressure of the deposition chamber was below 2.7 × 10^−5^ mbar using a diffusion pump backed by a rotary vane pump, and the working pressure was kept constant at 2.6 × 10^−2^ mbar. Then, the contamination on the target surface was removed by pre-sputtering for 10 min with a sputtering power of 45 W. Before deposition, the substrate was preheated at 300 °C for 15 min using a halogen lamp placed under the substrate holder. The thinnest films were carried out with a deposition time of 2 min while the thicker films were deposited with the multiple times of 10 min, i.e., 3 × 10, 6 × 10 and 12 × 10 min or 30, 60 and 120 min. The deposition of multiple times gives the film relaxation and atom interdiffusion that can reduce film cracking and reduce peeling off. For the deposition times of 30, 60 and 120 min, Sb_2_Te_3_ film was deposited for 10 min and was baked at 400 °C for 5 min. This deposition process was repeated until it reached the deposition times 30, 60 and 120 min. To improve the structural and thermoelectric properties, all films were post-annealed under an Ar atmosphere at 350 °C for 30 min as shown in Figure 1.

### 2.2. Materials Characterization

Energy dispersive X-ray spectroscopy (EDX, Oxford Instruments X-Max 20, Singapore) was used to measure the average atomic percentage of Te on the surface over a large area. The orientation plane of the Sb_2_Te_3_ films was characterized by X-ray diffraction (XRD D-MAX, Rigaku Corp., Japan) using CuKα source, and the surface morphology and thickness were determined with field emission scanning electron microscopy (SEM, EVO MA10, Carl Zeiss Microscopy GmbH, Jena, Germany). The electrical transport properties were measured with the Van der Pauw method by Hall effect measurements at room temperature (Ecopia, HMS 3000, Ecopia Corp., South Korea). Finally, the Seebeck coefficient and electrical conductivity were measured using a ZEM-3 apparatus (ULVAC-RIKO Inc., Yogohama, Japan) at 50–300 °C.

The output performance of a single planar Sb_2_Te_3_ film with different thicknesses was obtained. In the measurement, the deposited films were cut into 1.0 cm × 0.2 cm and tested by placing them on the heating platform, and the other sides were placed in the cold state to determine the temperature difference within the single planar thermoelectric field. The heat travelled through the thermoelectric planar between the hot and cold junction, and the carriers were also transported. As the device reached a steady state, short circuit current and open circuit voltage measurements were taken by a digital multimeter (Keithley 2100). At the same time, the temperature difference between both sides of a single plane was measured by a thermal infrared camera. The schematic setup can be seen in Figure 2.

## 3. Results and Discussion

The Sb_2_Te_3_ film thicknesses with various deposition times are shown in Table 1 as monitored by cross-sectional FE-SEM images. It is seen that the thickness increased as the deposition time was increased. The growth rate of all films was approximately 0.15 µm/min. In this work, the film thicknesses were found to be 0.3, 4.5, 9.0 and 16.7 µm for the deposition times of 2, 30, 60 and 120 min, respectively. The structures of the films were characterized by XRD and the results are shown in Figure 3.

In Figure 3, all films can be indexed to rhombohedral crystal structures (JCPDS card no. 15-0874, space group of $R\;\bar{3}\;m$) [15], which confirms the Sb_2_Te_3_ crystalline structure. In comparison with the lowest thickness, there are evident broad diffraction peaks. The broad peak should be attributed to the nanoscales of the film [16]. The orientation planes (009) and (0015) arose at the lowest thickness with a relatively high substrate temperature during deposition. The orientation plane along (*00l*) was observed, indicating that the atomic interactions between the Sb_2_Te_3_ atoms and the substrate can obstruct nucleation in the vertical direction [17,18]. As the film thickness increased to 4.5, 9.0 and 16.7 µm, the peak positions and their intensities changed and became sharper. The preferred orientation of the (015), (1010), (110) and (205) lattice planes was observed, which indicates that the atoms of the thick layer cannot obtain enough energy for lateral movement on the surface, leading to random growth in the direction vertical to the substrate [18,19]. During growth to the designation layer, the residual stress generally accumulates as the deposition time was increased [20]. Furthermore, the uneven distribution of thermal energy of the thick layer affects the growth process, leading to the surface diffusion of adatoms to grow in the ab-plane [18,19]. All the results suggest that all films are formed with a high crystallinity in the (015) plane. The sharper and intense diffraction peaks demonstrate the grain refinement along with the large strain [1].

The crystallite size, lattice strain and dislocation density are examined by Debye-Scherer’s equation.
(1)$$D=\frac{k\lambda }{\beta cos\theta }$$
(2)$$\varepsilon =\frac{\beta cos\theta }{4}$$
(3)$$\delta =\frac{1}{D^{2}}$$
where *D* is the calculated crystallite size, *λ* is the X-ray wavelength (Cu Kα = 0.154 nm), β is the full width at half maximum (FWHM) and *θ* is the Bragg diffraction angle. The calculated crystallite size, dislocation and lattice strain are presented in Table 1. The crystallite sizes at the (015) plane of the deposited Sb_2_Te_3_ films are N/A, 43, 40 and 18 nm for film thicknesses of 0.3, 4.5, 9.0 and 16.7 µm, respectively. It has been observed that the FWHM increases with increasing film thickness from 4.5 to 16.7 µm, and thus, the average crystallite size decreases. These results indicated that the crystallinity of the deposited film decreased as the film thickness increased. Consequently, the increase in the lattice strain (ε) and dislocation (δ) of the deposited films as a function of increasing the film thickness may be due to lattice mismatch and a change in the imperfection inside the film during the growth process [12,21].

EDS provides the average atomic percentage of Te for all annealed Sb_2_Te_3_ films, as shown in Table 1. The percentage of Te indicated that all samples were close to the stoichiometric ratio of 2:3 (Sb:Te). A small stoichiometric change can be obtained due to the thermal treatment process, which is caused by the evaporation of Te, considering the vapour pressures of constituent elements [18,22].

Figure 4 shows the surface morphologies of annealed Sb_2_Te_3_ films as a function of film thickness. It is obvious that plate structures and smooth surfaces are obtained with a film thickness of 0.3 µm, as seen in the red circle of Figure 4a. The high magnification surface SEM image illustrates the flat round grains corresponding to the texture of 00l orientation. The deposited film has a layered microstructure, indicating that the grains grow in the direction parallel to the substrate, which agrees with the XRD results. When the film thickness increased to 4.5, 9.0 and 16.7 µm, an ordinary structure was obtained. As the film thickness increases, the deposited atoms do not have enough energy for lateral movement on the surface, leading to random growth. In addition, a rougher surface with a small compact grain was clearly observed at a film thickness of 16.7 µm, as seen in Figure 4d.

Figure 5 shows a schematic of the growth process of the deposited Sb_2_Te_3_ films with different thicknesses. All films were grown under the same conditions with different deposition times to determine the effect of the designated films on the thermoelectric properties. During the deposition of the film, the thermal treatment process was repeated to reach the designated deposition thickness, as shown in Figure 1. In the sputtering process, the deposited atoms diffused on the surface of the substrate by giving up their kinetic energy and thermal diffusion. The mobility of the deposited atoms is also related to the appearance of the microstructure and morphology [17,23]. The diffusion and agglomeration between the grains can be yielded by thermal treatment. In the case of a relatively low thickness, a relatively high substrate temperature leads to a higher diffusion of the deposited atoms. The interaction between atoms is smaller than their bonding to the substrate. A two-dimensional layered structure parallel to the substrate is obtained. As the film thickness increased above 0.3 µm, the effect of substrate temperature decreased, leading to a reduction in the surface mobility of the deposited atoms. As the thickness of the film increases, the interaction between its atoms is stronger than the interaction with the substrate surface, causing the atoms to bond strongly to each other and grow into many three-dimensional structures. The transition from a layered structure to island growth occurs at a critical layer thickness, which is dependent on physical properties such as surface energy and lattice parameters. It is clarified that the film thickness and the substrate temperature significantly contribute to the growth process. After increasing the film thickness to a relatively high value of 16.7 µm, a small grain occurred. It may be that thermal energy during growth of the thick layer is not enough to diffuse and agglomerate the deposited atoms on the surface of the film [23], leading to the formation of a small compact grain on the film surface.

The Hall measurements confirmed that the majority of the carriers of the deposited films were holes. The carrier concentration and mobility measurements at room temperature are shown in Table 1. The carrier concentrations are 8.8 × 10^19^, 4.1 × 10^19^, 5.6 × 10^19^ and 11.3 × 10^19^ cm^−3^ for film thicknesses of 0.3, 4.5, 9.0 and 16.7 µm, respectively. Typically, the carrier concentration of deposited films is sensitive to the chemical composition and dislocation density of the film during the growth process [12,24]. In this work, the chemical composition of all films was nearly stoichiometric. The Te contents slightly disappeared after the thermal process. The change in carrier concentration of the deposited films is quite dependent on the dislocation density inside the films [12]. From the XRD patterns, a broad peak is obtained, and the intensity diffraction peak tends to decrease while the film thickness is increased. As the film thickness increased, the lattice strain and dislocation density increased, as shown in Table 1. The carrier mobility normally depends on the scattering dominance and carrier concentrations. A relatively high mobility can have a moderate carrier concentration [25]. Moreover, the grains of thick films will grow larger than those of a thin layer, resulting in a reduction in grain boundary scattering. Thus, the mobility of deposited films with a large thickness is relatively increased. Upon further increasing the film thickness to 9.0 µm, the mobility slightly decreased, resulting in a rise in the carrier concentration. In addition, a rougher surface with a small compact grain was obtained as the film thickness was increased to 16.7 µm, leading to a decrease in electric mobility.

The variation in electrical resistivity as a function of film thickness of the Sb_2_Te_3_ films is exhibited in Table 1. The results indicated that the electrical resistivity decreased as the film became a thin layer. Thin layered structure growth along the in-plane direction (*00l*) orientation provides a preferential method for carrier transport along the ab-plane and leads to an increase in carrier mobility and reduction in electron resistivity [26]. In cases of increasing the film thickness, the electrical resistivities are 2.16 × 10^−3^, 1.50 × 10^−3^ and 1.31 × 10^−3^ Ω.cm, which decrease with increasing film thickness of 4.5, 9.0 and 16.7 µm, respectively. It is well known that the resistivity is determined by the expression:ρ=1neμ, where *n* is the carrier concentration, *e* is the charge unit and μ is the mobility. The decrease in resistivity is mainly caused by the increase in the carrier concentration, leading to a rise in electrical resistivity in the films.

The carrier concentration and Seebeck coefficient variations as a function of film thickness are exhibited in Figure 6. The positive Seebeck coefficient of all deposited films indicates p-type behaviour, related to Hall measurements. At near room temperature, the Seebeck coefficient for the film thickness of 0.3 µm was found to be 131 μV/K. The *S* values rapidly increased as the thick layer increased to 4.5 µm and reached a maximum of 219 μV/K. For film thicknesses above 4.5 µm, the *S* values slightly decrease linearly. The variation in *S* values as a function of film thickness is clearly inversely related to the carrier concentration. The relationship between the Seebeck coefficient and the carrier concentration can be expressed by the Mott relation in Equation (4), where $\kappa_{B},h,m^{*},n,T$ are Boltzmann’s constant, Plank’s constant, effective mass of the charge carrier, carrier concentration and absolute temperature, respectively. The Seebeck coefficient is a factor that reflects the entropy transported per charge carrier and thus decreases as the carrier concentration increases [27].
(4)S(T)=8π2κB23eh2m∗Tπ3n23

The temperature dependence of the electrical conductivity of all deposited Sb_2_Te_3_ films with different film thicknesses is shown in Figure 7a. The electrical conductivity is slightly decreased when the applied temperature is increased in the range of 50–300 °C, which indicates typical metallic transport behaviour. When the applied temperature is increased, the carrier transport significantly interacts with the impurity inside the film, causing the mobility to decrease proportionally to T^−3/2^ [25,28]. The temperature dependence of the Seebeck coefficient shows an increasing trend in performance as the applied temperature is increased, as seen in Figure 7b. In theory, the Seebeck coefficient directly depends on the temperature and inversely depends on the carrier concentration [29]. The maximum Seebeck coefficient was obtained as 240 μV/K at 300 °C with a film thickness of 4.5 µm. When the film thickness is above 4.5 µm, the Seebeck coefficient slightly decreases. The electrical conductivity and the Seebeck coefficient of the deposited films were calculated to obtain the thermoelectric power factor: $PF=S^{2}\sigma$. The temperature dependence of the power factor as a function of film thickness is shown in Figure 7c. The power factor of the layered structure at room temperature is less than 1.5 mW/mK^2^. As the film thickness increases to 9.0 µm, the maximum power factor is obtained as 2.6 mW/mK^2^ due to the enhancement of the electrical conductivity and the Seebeck coefficient. In the case of the variation of the applied temperature, the deposited Sb_2_Te_3_ films exhibit optimum thermoelectric properties, and the maximum power factor is raised to 2.73 mW/mK^2^, observed for a film thickness of 9.0 µm with increasing applied temperature up to 100 °C.

A comparison of the thermoelectric properties of the deposited Sb_2_Te_3_ thick film with a thickness of a few micrometers using various methods is shown in Table 2. Although this method is a complex and long-term process, the obtained power factor is quite high when compared with other methods. Thus, it is evident that the selected method with an optimized growth process, such as an increase in the film thickness and a suitable thermal treatment process, is important to develop high efficiency deposited Sb_2_Te_3_ thick films.

The thermoelectric output power of the planar Sb_2_Te_3_ films was then calculated based on the voltage and current as a function of the temperature difference $\left(\Delta T\right)$. The electrical voltage $\left(V_{oc}\right)$ and output power $\left(P_{out}\right)$ versus temperature difference in thermoelectric single planar films with various film thicknesses were measured, as shown in Figure 8. After applying the temperature difference across the thermoelectric element, it can be seen that the deposited thin film has a relatively low voltage, leading to a small thermoelectric power. As the film thickness increases, the cross-sectional area of the element increases, and a high voltage and output power are obtained [29]. As a result, $V_{oc}$ increases linearly as a function of the applied temperature difference. In this work, the highest value of the planar thermoelectric leg, with a thickness of 9.0 μm at a deposition time of 60 min, is 0.75 mV at ΔT = 58 K, and the maximum $P_{out}$ of 0.032 µW is also obtained. For a thickness of 16.7 μm with a deposition time of 120 min, $V_{oc}$ and $P_{out}$ were slightly decreased, which may be due to the reduction of the film properties during growth, resulting in the decrease in the power factor. This result indicated that increasing the cross-sectional area as the film thickness of the element increased partially affected the output efficiency.

## 4. Conclusions

Designated p-type Sb_2_Te_3_ films with different thicknesses with varying deposition times of 2, 30, 60 and 120 min were successfully prepared on flexible substrates via heat treatment-assisted DC magnetron sputtering. The structure revealed that the preferred orientation plane along the (*00l*) plane was changed to the (015) plane as the thickness of the film increased, indicating that the atomic interactions between atoms and substrate can obstruct nucleation in the vertical direction. This texture change had a significant impact on electrical transportation. The residual stress typically accumulates as the thick layer increases, which is related to the diffusion of deposited atoms on the substrate surface and to the appearance of microstructure and morphology. The thermal energy of the thick dimension is not enough cause diffusion and agglomeration. The resultant surface morphology showed small compact grains as the film thickness increased. All imperfections inside the films are affected by the thermoelectric properties. The obtained maximum power output factor is 2.73 mW/mK^2^ for a film thickness of 9.0 µm at an applied temperature of 100 °C. Experimental characterization showed that increasing the film thickness to 9.0 μm at a deposition time of 60 min can produce an output power of 0.032 μW (0.75 mV) at a temperature difference of 58 K. The increase in the designated thickness of the planar single leg (cross-sectional area) affects the output efficiency. This indicated that the output power of planar single-leg Sb_2_Te_3_ films from nanowatts to submicrowatts can be useful in terms of energy-harvesting abilities.

## Figures and Tables

**Figure 1 materials-15-08850-f001:**
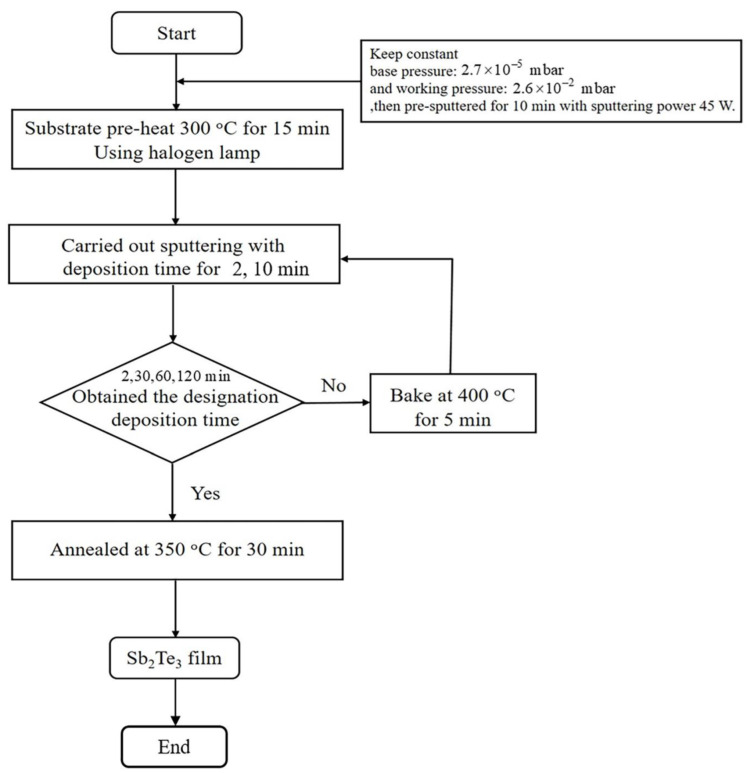
Schematic of the deposition method.

**Figure 2 materials-15-08850-f002:**
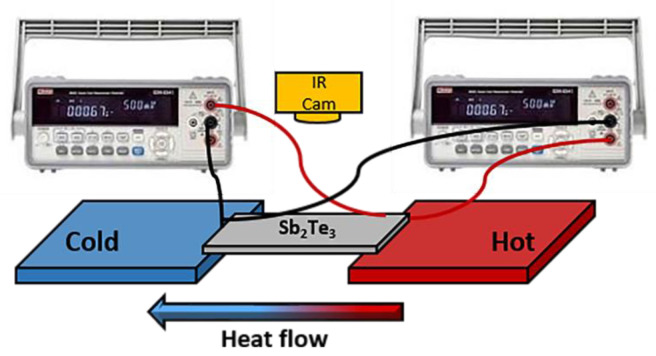
Experimental setup for output performance measurement of a single planar Sb_2_Te_3_ film.

**Figure 3 materials-15-08850-f003:**
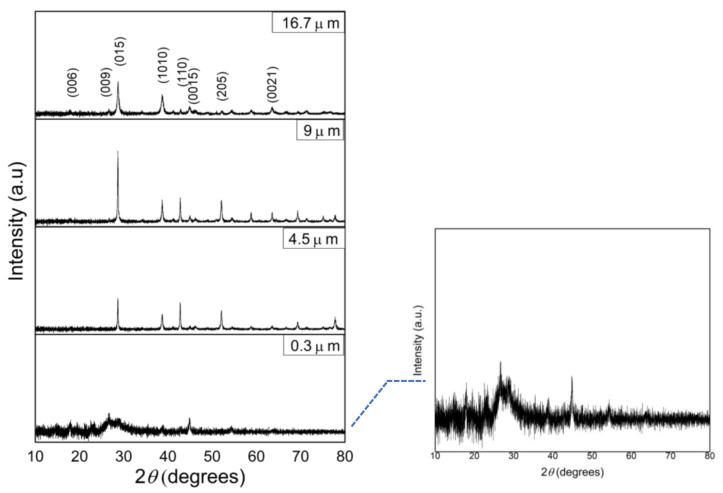
XRD patterns of deposited Sb_2_Te_3_ films with different film thicknesses. The inset shows the rescaled pattern of film thickness of 0.3 µm.

**Figure 4 materials-15-08850-f004:**
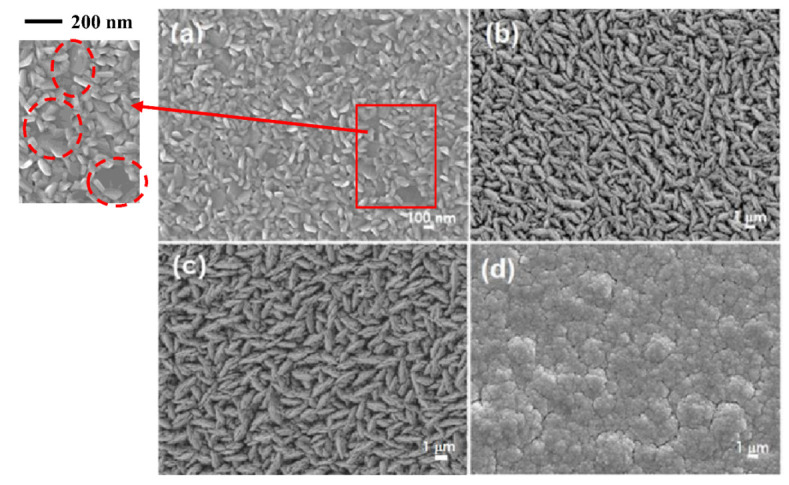
Surface morphology of annealed Sb_2_Te_3_ films with different film thicknesses: (**a**) 0.3, (**b**) 4.5, (**c**) 9.0 and (**d**) 16.7 µm.

**Figure 5 materials-15-08850-f005:**
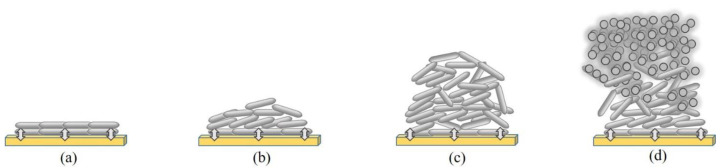
Schematic of the growth process of the deposited Sb_2_Te_3_ films with different film thicknesses: (**a**) 0.3, (**b**) 4.5, (**c**) 9.0 and (**d**) 16.7 µm.

**Figure 6 materials-15-08850-f006:**
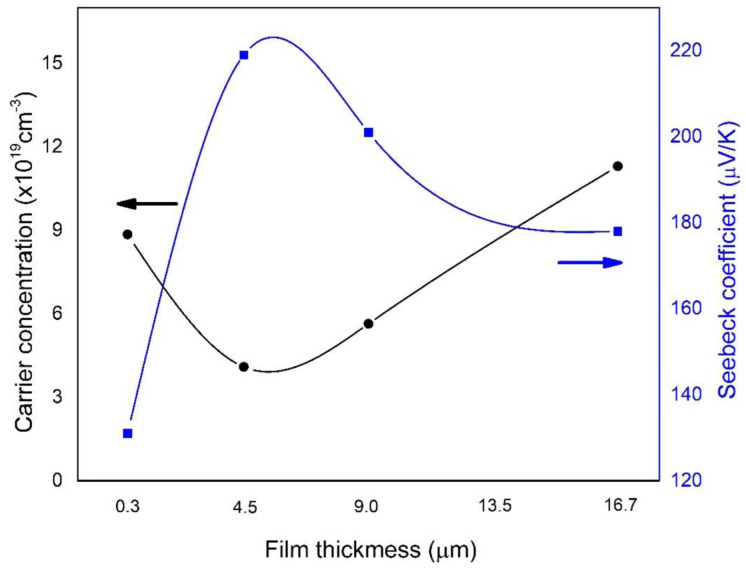
Carrier concentration and Seebeck coefficient of deposited Sb_2_Te_3_ films as a function of film thickness.

**Figure 7 materials-15-08850-f007:**
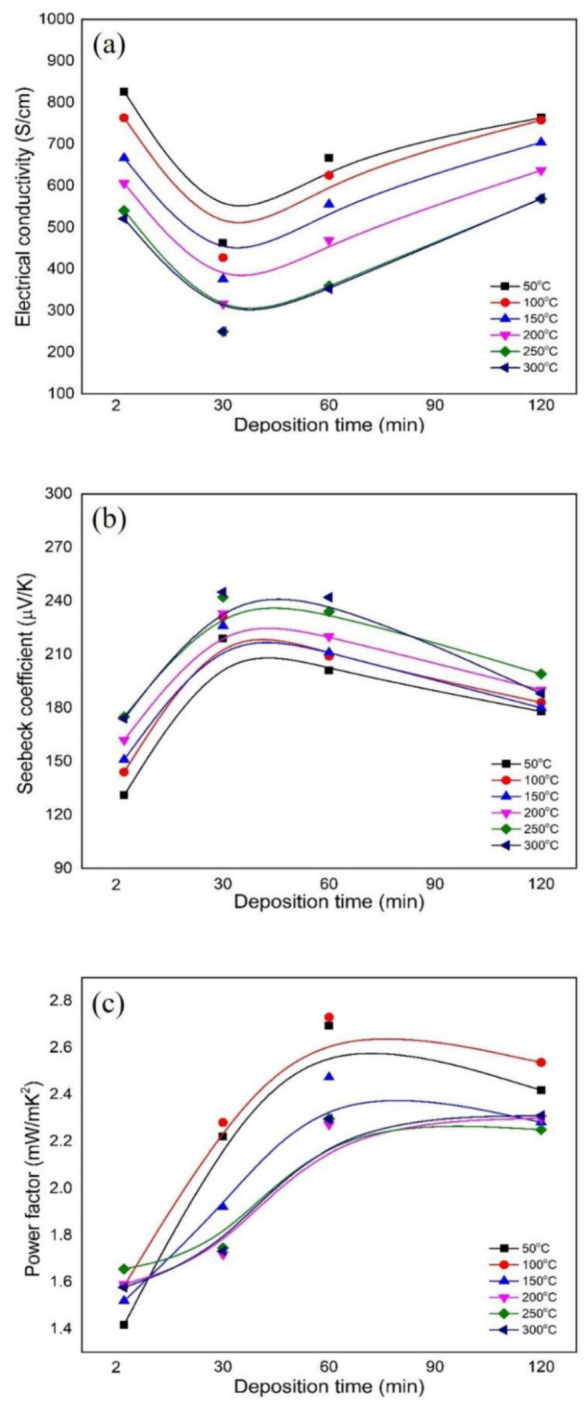
Temperature dependence of the thermoelectric properties as a function of film thickness of the deposited Sb_2_Te_3_ films: (**a**) electrical conductivity, (**b**) Seebeck coefficient and (**c**) power factor.

**Figure 8 materials-15-08850-f008:**
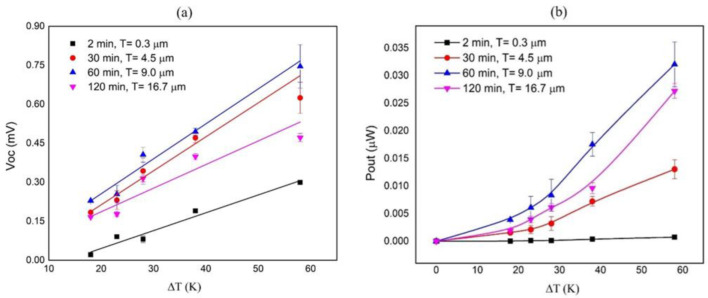
Temperature dependence of (**a**) Open circuit voltage and (**b**) Output power of single planar Sb_2_Te_3_ films with different film thicknesses.

**Table 1 materials-15-08850-t001:** Thickness, percentage of Te, lattice stain and crystalline size of deposited Sb_2_Te_3_ films with different film thicknesses.

Thickness (μm)	%Te	Crystalline Size (D: nm)	Dislocation (*δ* × 10^14^ Lines/m^2^)	Strain (ε × 10^−3^ Line^−2^ m^−4^)	Carrier Concentration (×10^19^ cm^−3^)	Mobility (cm^2^/Vs)	Resistivity (×10^−3^ *Ω*.cm)
0.3	58.73	N/A	N/A	N/A	8.9	58	1.21
4.5	58.11	42.6	5.4	0.81	4.1	75	2.16
9.0	58.09	40.3	6.2	0.86	5.6	68	1.50
16.7	58.51	18.3	30.8	1.89	11.3	42	1.31

**Table 2 materials-15-08850-t002:** Comparison of the thermoelectric properties of deposited Sb_2_Te_3_ films in this work and other studies.

Researchers	Method	Thickness (μm)	Resistivity (Ω.m)	PF (mW/mK^2^)
H.Shen [30]	co-evaporation	10.0	2.0 × 10^−5^	2.5
S.J. Kim [11]	screen printing	-	-	1.0
Z. Cao [6]	screen printing	3.5	5.0 × 10^−3^	0.2
O. Vigil-Galan [31]	close space vapour	15.3	3.0 × 10^−5^	0.4
M. Mizoshiri [8]	thermal-assisted DC sputtering	100.0	2.0 × 10^−5^	1.6
This work	DC sputtering	8.9	1.5 × 10^−5^	2.7

## Data Availability

Not applicable.

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
