# Peer review of "Thickness Dependence of Thermoelectric Properties and Maximum Output Power of Single Planar Sb2Te3 Films"

_materials, 2022, doi:10.3390/ma15248850_

Round 1
Reviewer 1 Report
The authors here are focusing on a topic relevant to the journal and of value to the readers. In this field, it is necessary for researchers to come up with new processes to further improve the thermoelectric performance of the films. Since this is a process development focused paper, I believe it is critical to add in control experiments that confirm the actual need for the multiple steps of deposition and heat-treatment. Although the power factor does show good improvement from previous works, more work is needed to conclude the following: "Thus, it is evident that the selected method with an optimized growth process, such as an increase in the film thickness and a suitable thermal treatment process, is important to develop high efficiency deposited Sb2Te3 thick films.". It is critical for the readers to learn how was the process developed to get to the "optimized conditions". Kindly consider showing trends of heat-treatment temperature, time, number of steps, etc., to show the optimized conditions. Some of the following questions further explain the above comments:
1. The experimental procedure details and the Schematic 1 is not valid for the 2min deposition time process, for 0.3um film. This needs to be corrected. How was the thinnest film deposited? Were the thicker films deposited as multiples of 2min or 10min of deposition?
2. Instead of depositing 10min then heat-treatment, why is the thicker film not deposited at once, and then heat-treated only once? This should be a control experiment to compare with a 120min film, which was deposited and heat-treated in 12 steps. How are the film properties, grain size, stress etc. different in a 12-step film versus a direct 16um thick film heat-treated only once? This heat-treatment could either be only 1x the time or 12x the time to compare the effects. Similar experiments must be carried out for each thickness, to confirm the thickness-dependence of the properties. Without these experiments, the following conclusion cannot be made "It is clarified that the film thickness and the substrate temperature significantly contribute to the growth process."
3. Must show the cross-section SEM images. This will also help understand the growth thickness model described in Figure 5. Also, without the cross-section SEM, the sentence "The deposited film has a layered microstructure, indicating that the grains grow in the direction parallel to the substrate, which agrees with the XRD results." cannot be confirmed.
4. "A two-dimensional layered structure parallel to the substrate is obtained. As the film thickness increased to 4.5 and 9.0 µm,..." I strongly recommend that films with thickness between 0.3um and 4.5um must be included to complete the analysis. It does seem like there will be a transition from 2-D to 3-D layering of the grains. Also, authors have previously shown in https://doi.org/10.1155/2019/6954918 that the thickness in the range of sub-microns is critical. Without comparing the films in the 0.3-4.5um range, the following claim cannot be confirmed "The transition from a layered structure to island growth occurs at a critical layer thickness, which is dependent on physical properties such as surface energy and lattice parameters. "
5. Many sentences in the Introduction and Experimental Procedure sections are not clear and references are incorrect. Kindly confirm the relevance and accuracy of each reference and sentences. For instance, in Section 1 "Several techniques have been deposited on flexible Sb2Te3 films". Or in Section 2.1 "First, polyimide (DuPont Kapton) substrates were prepared and ultrasonically cleaned, as described in our previous work [13-14]." Here, the references are incorrect. This work is not from the authors. I think the following are the correct references: https://doi.org/10.1155/2019/6954918 and https://doi.org/10.1007/s11664-017-5303-5
Author Response
Thank you for your comments. I have already edited manuscript as your comments.

Reviewer 2 Report
The manuscript “Thickness dependence of thermoelectric properties and maximum
power output of a single Sb2Te3 leg” was indicated that the maximum power factor is 2.73 mW/mK2 obtained with a film thickness of 9.0 μm and an applied temperature of 100 °C. Planar Sb2Te3 produced a maximum output power of 0.032 mW for a temperature difference of 58 K.The P-type Sb2Te3 films with different thicknesses were deposited on flexible substrates by using of DC magnetron sputtering. In my opinion, the authors have a complete discussion of experiment and result analysis in this manuscript. This can make it easy for readers to understand the content and future work of related research. Although the results appear to be interesting, reliable and convincing in this manuscript, I think that there are also little issues as follow, and this manuscript can be accepted and published in this journal after revised.
1. The English and frame of paper should be revised. In parts it is not clear what is meant by the text.
2. I think that the title and keyword should be revised and more acerated with the research content. Such as Seebeck effect etc..
3. In Fig. 1, the flowchart form and text should be represented by formal graphs, such as "start" and "end”.
4. In table 1, the form of table should be revised, such as the horizontal lines in the table should be hidden.
5. In Fig. 4, The surface morphology of Sb2Te3 film looks like the surface with nano (leg) structure. The author should have a consistent name to represent this plane before and after of the manuscription.
6. In Figs. 1-8, the resolution of the figure, the text should be enlarged and the lines should be thickened to make it easy for the reader to read.

Author Response

(The authors gave the same response as above.)

Round 2
Reviewer 1 Report
I see the authors have made some changes to the manuscript as per the suggestions from the reviewers. Also, in the cover letter, there are some additional data presented, for example, the cross-section SEM image. I'd suggest authors to add that data in the manuscript, as it will be helpful for the readers to understand the Figure 5 schematic. However, the cross-section SEM images do not fully match the simple schematic drawn by the authors. Kindly clarify your findings in paper.
The explanation regarding the film cracking and peeling off when deposited in a long single step would be of interest to the readers who would follow this paper results to replicate the work. If you add 1-2 sentences regarding these findings in the experimental section, that will greatly serve the community.
There are several sentences in the Introduction that need extensive English editing. For example, "Several techniques have been deposited on flexible Sb2Te3 films."
Additional minor edits:
1. Section 2.2 line 3-4, XRD Rigaku Model ... Some text here is not in English. Kindly correct the typo.
2. Table 1 column name still reads as "Crystalline size" instead of crystallite size, which has been corrected at various places in the manuscript.
Author Response
The revised manuscript was edited as reviewer comment.
